# Lancastrians, Tudors, and World War II: British and German Historical Films as Propaganda, 1933–1945

**William B. Robison** 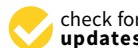

Department of History and Political Science, Southeastern Louisiana University, Hammond, LA 70402, USA; william.robison@southeastern.edu

**Abstract:** In World War II the Allies and Axis deployed propaganda in myriad forms, among which cinema was especially important in arousing patriotism and boosting morale. Britain and Germany made propaganda films from Hitler's rise to power in 1933 to the war's end in 1945, most commonly documentaries, historical films, and after 1939, fictional films about the ongoing conflict. Curiously, the historical films included several about fifteenth and sixteenth century England. In *The Private Life of Henry VIII* (1933), director Alexander Korda—an admirer of Winston Churchill and opponent of appeasement—emphasizes the need for a strong navy to defend Tudor England against the 'German' Charles V. The same theme appears with Philip II of Spain as an analog for Hitler in Arthur B. Wood's *Drake of England* (1935), William Howard's *Fire Over England* (1937), parts of which reappear in the propaganda film *The Lion Has Wings* (1939), and the pro-British American film *The Sea Hawk* (1940). Meanwhile, two German films little known to present-day English language viewers turned the tables with English villains. In Gustav Ucicky's *Das Mädchen Johanna* (*Joan of Arc*, 1935), Joan is the female embodiment of Hitler and wages heroic warfare against the English. In Carl Froelich's *Das Herz der Königin* (*The Heart of a Queen*, 1940), Elizabeth I is an analog for an imperialistic Churchill and Mary, Queen of Scots an avatar of German virtues. Finally, to boost British morale on D-Day at Churchill's behest, Laurence Olivier directed a masterly film version of William Shakespeare's *Henry V* (1944), edited to emphasize the king's virtues and courage, as in the St. Crispin's Day speech with its "We few, we proud, we band of brothers". This essay examines the aesthetic appeal, the historical accuracy, and the presentist propaganda in such films.

**Keywords:** Britain; film; Germany; Lancastrian; Nazi; Tudor; World War II

---

> In this hard material age of brutal force, we ought indeed to cherish the arts.
>
> Winston Churchill, 1938

> Only in war do the muses realize their full potential. Art is not a form of peacetime divertissement, but a sharp psychological weapon.
>
> Joseph Goebbels, 1939

## 1. Introduction

Great Britain and Germany were at war longer than any other combatants in World War II, from 3 September 1939 to 8 May 1945. However, the film industries in both nations were busy from Adolf Hitler's appointment as Chancellor of Germany on 30 January 1933 to war's end, releasing hundreds of comedies, dramas, and musicals, many intended to distract citizens from the fear of approaching conflict and then from the hardships of war. These films often reflect their makers' ideological leanings. However, both nations also deliberately produced propaganda films, for example, such documentaries as Leni Riefenstahl's *Triumph des Willens* (*Triumph of the Will*, 1935) and J. B. Priestley's *Britain at Bay*

(1940) and, once hostilities began in 1939, fictional films about the ongoing conflict like the British *49th Parallel* (1941) and the German *Die Große Liebe* (*The Great Love*, 1942). Historical films also served a propagandistic purpose. This essay focuses on seven historical films that are distinctive because they are based on English history in the Lancastrian and Tudor periods. Although spawned by Anglo-German hostility, only one features German characters, and they are non-adversarial—Anne of Cleves and company. Otherwise, France, Scotland, and Spain represent Germany.[1]

Four are British. In 1933, Alexander Korda, an admirer of Winston Churchill and opponent of appeasement, directed *The Private Life of Henry VIII*, wherein the king calls for a stronger navy to defend England against the 'German' Emperor Charles V, who serves as an analog for Hitler but was actually Burgundian. In 1935 Arthur B. Woods directed *Drake of England* (a.k.a. *Drake the Pirate*), in which Sir Francis Drake convinces Elizabeth that Philip II of Spain, another Hitler analog, is a greater threat than her appeasing courtiers realize. In 1937, Korda produced and William Howard directed *Fire Over England*, which again features Philip as a stand-in for the Führer and culminates in 1588 with England's defeat of the Spanish Armada and Elizabeth's Tilbury Speech, both of which reappear in Korda's 1939 documentary *The Lion Has Wings*. In 1944, at Churchill's behest and to boost British morale for D-Day, Laurence Olivier directed and starred in a film version of Shakespeare's *Henry V* that highlighted the king's virtues and courage. A 1940 American film directed by Michael Curtiz, *The Sea Hawk*, is included because it encouraged pro-British feeling and had yet another Hitlerian Philip. Two films are German. In *Das Mädchen Johanna* (*Joan of Arc*, 1935), directed by Gustav Ucicky, the Maid of Orléans is a different sort of analog for Hitler, one who is favorable and female. *Das Herz der Königin* (*The Heart of a Queen*, 1940), directed by Carl Froelich, has a Germanized Mary, Queen of Scots opposed to a decidedly Churchillian Elizabeth I.[2]

Clearly this essay is narrow in scope. It reflects the interests and limitations of its author, whose expertise is Tudor politics, the English Reformation, and films about fifteenth and sixteenth century England. Thus, it is not particularly concerned with film theory or other aspects of film studies. It does not participate in the larger agendas of scholars whose work deals with the whole body of British or German film or even all historical films in either country c.1933–1945, for example, the post-colonial critique of imperialism in British cinema or the prioritization of artistic over propagandistic elements in the study of German films. While this study notes the varying degree to which the selected films undermined their own propagandistic message, it devotes little attention to the persistence of elements of Weimar cinema in Nazi films or the influence of Hollywood on British or German filmmakers. The aforementioned matters are best left to the appropriate experts. Thus, while this essay places the seven selected films and their directors, producers, and actors into a larger context, it is primarily about why, how, and to what effect certain state-supported British filmmakers and their state-controlled German counterparts manipulated historical memory to either exploit or undermine the heroic proto-nationalistic legacy of the Lancastrians and Tudors for propagandistic purposes.

That all seven films involve related subjects allows fruitful comparison of their aesthetic appeal, historicity, and propaganda-related presentism. This study proceeds on the assumption that aesthetic and propagandistic appeal are related, that a well-made film is more likely to be more persuasive, and that the right message can make a movie more acceptable to a certain audience. It deems consideration of the historicity of films not only worthwhile but essential, and rejects the postmodern claim that film is a 'mode of history' no more 'fictional' or 'fictive' than written history. Therefore, it must be said at the outset that none of these films are historically accurate. Though they deal mostly with real historical persons, circumstances, and events, all are highly fictionalized. That is hardly surprising, for the makers of historical films are rarely much concerned with historicity. Their movies

---

[1] Most films mentioned are available on Blu-Ray or DVD, most British individuals are in the *Oxford Dictionary of National Biography*, https://www.oxforddnb.com/, and most German individuals are in (Bock 2009).

[2] The seven films are available as (*Das Herz der Königin* 2006; *Das Mädchen Johanna* 1935; *Drake of England* 2014; *Fire Over England* 1992; *Henry V* 2010; *The Private Life of Henry VIII* 2009; *The Sea Hawk* 2005).

usually tell us more about their own age than the one depicted onscreen, regardless of any agenda they may have, even if it is purely entertainment. However, propaganda is a special case, because evenhandedness runs counter to its purpose. Of course, the term has two meanings. Propaganda can have the relatively benign purpose of propagating information in support of a particular point of view, though even in that case it is hardly likely to be unbiased. However, it is often harmfully misleading, with propagandists distorting history in the service of brutal, dictatorial regimes.[3]

A case in point is Joseph Goebbels, who as Reich Minister for Propaganda oversaw filmmaking in Nazi Germany from 1933 to 1945. Like the Führer, he was fond of movies, and he took a close personal interest in their production. Under his supervision the state-controlled film company Ufa made several historical films that presented earlier German leaders as forerunners of the Führer, including Prussian rulers Frederick William I and Frederick the Great in *Der Alte und der Junge König* (*The Old and the Young King*, 1935), the unifier of Germany in *Bismarck* (1940), an anti-Semitic East African colonial leader in *Carl Peters* (1941), Frederick the Great in *Der Große König* (*The Great King*, 1942), Bismarck and Kaiser Wilhelm II in *Die Entlassung* (*The Dismissal*, 1942), and the anti-Semitic Viennese mayor Karl Lueger in *Wien 1910* (*Vienna 1910*, 1943). A few depicted Prussia and Britain as allies, i.e., *Kadetten* (*Cadets*, 1939) for the Seven Years War, *Der höhere Befehl* (*The Higher Command*, 1935), and *Kolberg* (1945) for the Napoleonic Wars. Others portrayed Britain in a negative light, such as *Ohm Krüger* (*Uncle Krüger*, 1941) about the German-backed South African Paul Krüger, who fought Britain in the Boer War; *Der Fuchs von Glenarvon* (*The Fox of Glenarvon*, 1940) and *Mein Leben für Irland* (*My Life for Ireland*, 1941) about Irish resistance; and *Titanic* (1943), which blamed Anglo-American capitalism for the 1912 maritime disaster. Gustav Ucicky's *Das Mädchen Johanna* and Carl Froelich's *Das Herz der Königin* reach back into history to suggest the English always have been cruel and greedy.[4]

Britain had no counterpart to Goebbels, though the British Ministry of Information sought to counter the work of the German Ministry of Propaganda, in part by making documentary films. However, Churchill encouraged filmmakers to produce propagandistic historical films both during his political exile in the 1930s and as prime minister during the war. He found capable allies in Korda and Olivier, who often worked together in the 1930s and 1940s. The Hungarian expatriate Korda arrived in Britain in 1932 after sojourns in Vienna, Berlin, Hollywood, and France, founded London Films, and directed or produced a variety of movies before war forced him back to Hollywood in 1940. He made romantic comedies, science fiction, documentaries, historical films like *The Rise of Catherine the Great* (1934) and *Rembrandt* (1936), propaganda films such as *Q Planes* (1939)—which featured Oliver—and *The Lion Has Wings* (1939), and quasi-historical films with nationalist elements including *Sanders of the River* (1936), *Dark Journey* (1937), *The Challenge* (1938), *The Rebel Son* (1939), and *The Four Feathers* (1939). After returning to America to complete *The Thief of Baghdad* (1940), Korda produced and directed *Lady Hamilton* (aka *That Hamilton Woman*, 1941), which starred Olivier and, though set during the Napoleonic Wars, consciously evoked Elizabethan success against the Spanish Armada. He addressed the Tudors more directly in *The Private Life of Henry VIII* and *Fire Over England*.[5]

Olivier began acting onstage in 1925 and onscreen in 1930. Interestingly, his first film role was in *The Temporary Widow*, a Weimar Republic era British-German collaboration directed by Gustav Ucicky. He became a classic leading man, and in addition to movies already mentioned, he appeared in several other productions by Korda's London Films, including the World War I drama *Moscow Nights* (1936), the documentary *Conquest of the Air* (1936) directed by Zoltan Korda, the romantic comedies *The Divorce of Lady X* (1938), *21 Days* (1940), and *49th Parallel* (1941), as well as the Ministry of Information documentaries *Words for Battle* (1941), *The Volunteer* (1943), and *Malta G.C.* (1943). Of the

---

[3]    On historicity (Carnes 1995; de Groot 2008; Freeman and Smith 2019; Hughes-Warrington 2006; Robison 2016a; Rosenstone 2018; Stubbs 2013); for a succinct statement of the postmodern view, see (Munslow 2007).

[4]    On Goebbels (Longerich 2015; Thacker 2009); on his involvement with film (Moeller 2001).

[5]    On the Ministry of Information (Grant 1994; McLaine 1979); on Korda (Korda 2007; Kulik 1990); an excellent documentary on Churchill and Korda is (*Churchill and the Movie Mogul* 2019); on *Lady Hamilton* (Cavell 2019).

filmmakers and actors discussed here, Olivier has achieved the greatest and longest-lasting fame, and his *Henry V* is the best known of the films this essay addresses and subject of the most scholarly analysis.[6]

## 2. History, Empire, and Legitimacy

It is natural that British filmmakers drew on fifteenth and sixteenth century English history. Henry V, Henry VIII, Elizabeth, and Drake are larger-than-life figures who fit the bill perfectly, particularly given how Britons remembered them in the 1930s and 1940s, not only in popular culture but also in the work of academic historians. The three monarchs were key figures in the Whig theory of history that remained influential prior to the post-war rise of Marxist historiography and other rival schools. Whig historians emphasized, sometimes in teleological fashion, the rise of the national state, development of constitutional government, growth of personal liberty, scientific progress, and emergence of Protestant liberalism, all linked to the Reformation and England's—later Britain's—triumphs over French and Spanish absolutism. Henry V was the victor of Agincourt, the epitome of the chivalric ideal, and in some respects the forefather of the Tudors. Henry VIII was the founder of the modern English state, the instigator of the English Reformation, and to some the father of the English navy. Elizabeth I was Good Queen Bess, Gloriana, the Virgin Queen married to her people, and the Protestant stalwart who survived Catholic plots centered on Mary, Queen of Scots and withstood the Spanish Armada. Drake was her most famous adventurer. All four seemed worthy of evocation in British propaganda.[7]

It might appear surprising that Germany produced films about the same period in history. The films in question certainly are little known in the English-speaking world today and have received comparatively little attention from scholars. Yet, Joan of Arc and Mary, Queen of Scots, both martyred at English hands, were excellent choices to undermine England's heroes and suggest modern Britain's liberal values were merely a hypocritical veneer obscuring a long history of brutality, intolerance, and greed. If Joan initially seems a highly improbable embodiment of Hitler, her Gallic ethnicity, the emasculating potential of her gender, her insultingly low social status, her victories over English troops who served Henry VI, her unjust and cruel execution, and her recent canonization in 1920 made her an ideal goad for Britain. Chances are the Nazis also enjoyed provoking their enemy France by appropriating a French saint to embody the Führer. Mary Stuart—half-French, half-Scots, doubly England's enemy—was a more obvious avatar of Teutonic virtues in contrast to a coldly imperialistic Elizabeth, who denied Mary's claim to the English throne and deprived her of freedom, the comforts of her Catholic faith, the loyalty of her son and heir James, and ultimately her life.

Legitimacy is an important theme in all seven films. Henry V, Joan of Arc, Henry VIII, Elizabeth I, and Mary, Queen of Scots all represented legitimacy but could easily be targeted as illegitimate. Competing claims to the English and French thrones ran throughout the dynastic struggles of the fifteenth and sixteenth centuries and provided ideal subject matter for the current regimes in Britain and Germany to claim legitimacy for themselves and to discredit the other. Both also implicitly or explicitly referenced empire for audiences familiar with imperial mythology. Britons and Germans reveled in believing their respective empires more ancient, venerable, and legitimate. The British and German films discussed here resonated strongly but differently with their respective audiences because they approached legitimacy from opposing points of view.[8]

---

[6]　Among the many biographies and critical studies, one the most accessible is (Ziegler 2013).

[7]　Hebert Butterfield originated the term 'Whig history' in (Butterfield 1965); examples of Whig historians include J. A. Froude, S. R. Gardiner, David Hume, James Macintosh, Thomas Babington Macaulay, J. E. Neale, A. F. Pollard, William Stubbs, and George M. Trevelyan.

[8]　For insights on the issue of legitimacy in Shakespeare's history plays and the reign of Elizabeth (Lake 2017; Greenblatt 2018); on the legitimacy of Henry VIII's children and the royal succession (Robison 2017); on Hitler's ability to project legitimacy (Ullrich 2016).

The British films underscored the legitimacy of the monarchy and state to elicit a patriotic response and an indignant reaction against challengers. The most obvious imperial symbols were Henry V, Henry VIII, Elizabeth, and Drake themselves. Henry V's invasion of France was an imperialistic gambit and created a cross-channel empire that might have surpassed its Angevin predecessor had he lived longer. Henry VIII not only revived Henry V's claim to the French throne and tried to emulate his military success, he declared himself a candidate for the throne of the Holy Roman Empire in the imperial election of 1519. Moreover, the conventional wisdom c.1933–1945, based on the now discredited argument of A. F. Pollard, was that he hoped to make himself emperor and put Cardinal Thomas Wolsey on the papal throne. During the English Reformation in the 1530s, Thomas Cromwell claimed England was an empire and Henry the equal of the Holy Roman Emperor and the pope. Henry also took the title King of Ireland in 1542 and with the Rough Wooing of 1542–1551 sought to unite the Tudor line with the Stuarts in Scotland through the marriage of Prince Edward to Mary, Queen of Scots. Elizabeth celebrated the defeat of the Spanish Armada in 1588 with a portrait depicting her with symbols of imperial might, notably a closed imperial crown and a globe on which she rests her hand. Historians long regarded 1588 as the turning point when a nascent English empire began to gain ground on a faltering Spanish Empire. Even now Britons emphasize the antiquity of the British Empire and compare it to its Roman antecedent.[9]

German films used Joan and Mary to assert the illegitimacy of British institutions, cast British imperialism in a negative light, and imply that Churchill was an upstart, and they did so for a German audience that regarded its own Empire as eminently respectable. The largely German Holy Roman Empire or First Reich originated in either 800 with Charlemagne or 962 with Otto the Great, cloaked its authority in the mantle of ancient Rome, fought for preeminence with the papacy, and lasted until 1806. Prussian Chancellor Otto von Bismarck created the Second Reich in 1871 (it was no coincidence that in 1877 Prime Minister Benjamin Disraeli got parliament to bestow the title of Empress of India upon Queen Victoria, whose daughter Vicky was married to Frederick, son of Kaiser Wilhelm). The German Empire lasted only until the end of World War I and the overthrow of Victoria's grandson Wilhelm II. However, Hitler intended his Third Reich, founded in 1933, to be another thousand-year empire and asserted its legitimacy and continuity with ancient Rome in contrast to the British Empire. Fortunately, he was to be disappointed.[10]

### 3. The Lancastrians

Here it will be useful to briefly survey Lancastrian and Tudor history, with emphasis on individuals who figure prominently in the seven films. The Anglo-French conflict that characterized the Lancastrian and Tudor periods and forms the backdrop for *Henry V* and *Das Mädchen Johanna* dated to 1066, when William the Conqueror seized the English throne while retaining the Duchy of Normandy in northwestern France, setting the stage for frequent wars between the Anglo-Norman kings and their French counterparts. Beginning in 1154, Henry II created an Angevin Empire that included England, Ireland, and roughly half of France, though between 1204 and 1214 John lost all but Gascony to Philip II, who made the French crown dominant in his realm. Intermittent warfare continued, however, and in 1295 France bolstered its position by entering into the Auld Alliance with Scotland, which lasted until 1560. In 1340—early in the Hundred Years War (1337–1453)—Edward III raised the stakes by asserting a claim to the French throne, though he renounced it in the 1360 Treaty of Brétigny, which made him Lord of Aquitaine. When fighting resumed in 1369, he reasserted the claim, and thereafter his successors continued to style themselves 'King of France' until George III finally renounced the title in 1800. Meanwhile, by 1380 the French Charles V had recovered all English-held territory except for

---

9    On Pollard (A. F. Pollard 1905, 1929); on empire (Elton 1982, pp. 338–77; Lehmberg 1970, chp. 8); on the Rough Wooing (Merriman 2000); on the Armada portrait (Strong 2003, pp. 131–33).
10   On the history of the three Reichs (Wilson 2016; Evans 2004).

the ports of Bayonne, Bordeaux, Brest, and Calais. The deaths of Edward III in 1377 and Charles V in 1380 left England and France in the hands of minors. In 1392 the French Charles VI went mad, leading to almost three decades of conflict over the regency between the Houses of Burgundy and Orleans (or Armagnac). In 1399 Henry, Duke of Lancaster usurped the English throne from Richard II and founded the Lancastrian dynasty as Henry IV; however, his uncertain title encouraged rebellion in England and Wales and threats from France and Scotland, which consumed his reign.[11]

Henry V was an ideal patriotic hero for stage and screen. Though William Shakespeare's *Henry IV*, *Parts 1 and 2* characterizes Prince Hal as a dissolute young man, in reality the Prince of Wales spent his father's reign fighting against the turncoat Percy family, whom he helped to defeat at the Battle of Shrewsbury in 1403, and Welsh rebels led by Owain Glyndwr, whereby he acquired great experience and extensive knowledge of strategy, tactics, and logistics. His father's declining health led him to take over the government of England between 1409 and 1411, but Henry IV resumed control during the last two years of his reign, during which father and son were often at odds. When he became king in 1413, Henry V moved quickly to legitimize the suspect Lancastrian monarchy, providing Richard II with a more honorable burial and restoring lands and titles to the heirs of rebels against his father. He foiled a potential rebellion in his own reign, the Southampton Plot of 1415. Both Shakespeare and Olivier ignore his important, if sometimes disturbing, role with regard to the Church, suppressing the Lollard heresy, having his one-time friend Sir John Oldcastle burned at the stake in 1417, and attending the Council of Constance, which restored the unity of Western Christendom after the Great Papal Schism and burned the Czech heretic Jan Hus.[12]

But, of course, it is his military exploits as king that figure in Shakespeare's *Henry V* and Olivier's film thereof. Henry epitomized the chivalrous warrior so beloved of the Middle Ages, in contrast to his troubled immediate predecessors, his French adversary Charles VI, and his weak successor Henry VI. He reignited England's hope for martial glory after an ignominious half-century and revived the Hundred Years War in 1415. His triumph at Agincourt that year is one of England's most famous, as well as a great set piece in British history, art, drama, and literature. It was the third great victory of English yeoman archers over French aristocrats after Crecy in 1346 and Poitiers in 1356, and allowed English commoners to share the glory then and vicariously in future generations. After allying with Philip Duke of Burgundy, Henry compelled Charles VI in the 1420 Treaty of Troyes to recognize him as heir to the French throne, give him the hand of his daughter Catherine of Valois, and declare his own son the Dauphin Charles a bastard. Henry died at his peak in 1422, just months before Charles VI, but he bequeathed both kingdoms to his infant son, who was crowned king of England in 1429 and of France in 1431. He also did not live long enough for his wars to become unduly burdensome or for his glory to fade. Instead, his reputation grew during Henry VI's long, faction-ridden reign and the Wars of the Roses, and in historical memory he remains the young hero, with chroniclers, poets, and playwrights having set the seal on his immortality.[13]

Joan of Arc, the title character in *Das Mädchen Johanna*, is renowned as the savior of France in the years after Henry's death. Certainly, her emergence in 1429–1431 inspired French armies to new fervor, but other factors played a larger role in the eventual French victory. Henry VI, a minor in Joan's day, grew to be an incompetent adult who suffered a severe mental breakdown in the 1450s, while the disinherited Dauphin, calling himself Charles VII, became a capable war leader. Henry's uncle John, Duke of Bedford, initially maintained English control over northern France with Burgundy's assistance, but political intrigue in England hampered him. Very importantly, in 1435 Bedford died and Burgundy renounced the English alliance. Subsequently, Henry lost everything but Calais by 1453,

[11]  For good surveys of the period 1066–1485 (D. Carpenter 2005; Rubin 2006); for more details see the relevant volumes in the Yale Monarchs Series, especially (Saul 1997; Given-Wilson 2016); on Shakespeare's history plays (Saccio 2000).
[12]  On Henry V (Allmand 1993; Curry 2018; Vale 2016).
[13]  Ibid.; on the Hundred Years War (Allmand 1988; Curry 2003); on Agincourt (Barker 2006; Curry 2015).

and his declining fortune helped trigger Cade's Rebellion in 1450 and the outbreak of the Wars of the Roses in 1455. Still, Joan was and remains a potent symbol.[14]

A peasant girl born c.1412 in Domremy, at age thirteen she began having visions of the Archangel Michael and other saints urging her to help Charles VII free France from English control. A popular prophecy said a woman would lose France and a virgin regain it, and Frenchmen believed this referred to Isabeau of Bavaria, who signed the Treaty of Troyes, and the virgin Joan. She browbeat Robert Baudricourt, commander of the Vaucouleurs garrison, into taking her to meet Charles at Chinon. He was skeptical, but his mother-in-law Yolande of Aragon persuaded him to send Joan with the army to Orleans after having her examined to ensure her moral purity. She helped raise the English siege of Orleans, followed by a series of victories that enabled Charles's coronation at Reims Cathedral, convincing the French that God had sent her and the English that she was a tool of the Devil. In 1430 she condemned the Hussite heretics and tried to organize a crusade against them in Bohemia. Later in 1430, the Burgundians captured Joan, Charles refused to ransom her, and they turned her over to the English, who had her tried and burned for witchcraft in 1431.[15]

However, in 1456, at Pope Callixtus III's behest, an inquisitorial court reexamined the evidence, found Joan innocent, and proclaimed her a martyr. She was not quiet in death. French troops made her sacrifice a rallying cry for the remainder of the Hundred Years War, the Catholic League did the same during the religious wars of the sixteenth and seventeenth centuries, Napoleon declared her a national symbol in 1803, Pius X beatified her in 1913, French troops sang songs about her in World War I, Benedict XV canonized her in 1920, and—especially interesting given Hitler's use of Joan—present-day radical right groups in France revere her. She is the subject of hundreds of films, operas, paintings, sculptures, and written works ranging across the last five centuries from Christine of Pizan's contemporary elegy to modern biographies, histories, poems, and novels, notably Mark Twain's *Personal Recollections of Joan of Arc* (1896).[16]

## 4. The Tudors

The star of *The Private Life of Henry VIII* is perhaps England's most famous king and a favorite of filmmakers, but he came to the throne by an unlikely path. In the 1450s, Richard, Duke of York emerged as a rival to Henry VI, and his son Edward IV seized the crown 1461 and in 1483 left it to his own minor son Edward V, who was ousted by his uncle Richard III (1483–1495). Meanwhile, Henry V's widow, Catherine de Valois, married Owen Tudor, and their son Edmund married Margaret of Beaufort, great-granddaughter of John of Gaunt. In turn, their son proclaimed himself 'the last Lancastrian', defeated Richard at Bosworth in 1485, and took the throne as Henry VII. He married Edward IV's daughter Elizabeth of York in 1486, uniting the Lancastrian and Yorkist lines with the Tudor dynasty. Among their children, Arthur died young; Margaret Tudor was the grandmother of both Mary, Queen of Scots and her husband Henry Stuart, Lord Darnley; Henry VIII fathered Mary I, Elizabeth I, and Edward VI; and Mary Tudor was the grandmother of Jane Grey.[17]

Henry VIII is best remembered today as the libidinous prodigy, alternately fun-loving and cantankerous, who married six times in pursuit of legitimate male heirs, fought the French, the Habsburgs, and the papacy, triggered the English Reformation, and became so fat he could not mount a horse without mechanical assistance. Both he and his hero Henry V enjoyed a legitimacy their usurping fathers were unable to achieve, treated heretics harshly, and sought glory on the battlefield in France. Henry VIII in 1509 married his brother Arthur's widow Catherine of Aragon, who gave birth to Mary; in 1533 divorced her and married Anne Boleyn, who gave birth to Elizabeth;

---

[14] On Henry VI (Wolfe 2001); on Cade's Rebellion (Harvey 1991); on the Wars of the Roses (C. Carpenter 1997; A. J. Pollard 2013).
[15] A good entry point to the voluminous literature on Joan is (Castor 2015).
[16] On Joan's influence (Fraioli 2002; Heimann and Coyle 2006; Meltzer 2001).
[17] On the Yorkists (Johnson 1988; Ross 1997, 2011); on the rise of the Tudors (Skidmore 2014); the standard biography of Henry VII is (Chrimes 1999), but see also (Penn 2012).

in 1536 had her executed for adultery and treason and married Jane Seymour, who gave birth to Edward and died in 1537; in 1540 married and divorced Anne of Cleves and married Catherine Howard, whom he had executed for adultery and treason in 1542; and in 1543 married Catherine Parr. He also broke away from the Roman Catholic Church in the course of seeking a divorce from his first wife, was excommunicated by Paul III in 1538, vacillated between an alliance with Francis I of France and one with the Holy Roman Emperor Charles V, faced the most serious rebellion of the Tudor period in the Pilgrimage of Grace (1536–37), made war on the continent with minimal success, and was willing to sacrifice to his ambition ministers, courtiers, and clergymen. Whig historians regard Henry as the founder of the modern state, though Geoffrey Elton credits Thomas Cromwell with the innovations in question, and more recent revisionists see modernization as a longer-term process. In Protestant historiography, Henry is the author of the English Reformation, though Catholic historians long disparaged him for his break with Rome, execution of Thomas More, dissolution of the monasteries, and other attacks on the Church. In propaganda terms, Henry can be the hero or the villain.[18]

Elizabeth is a major character in *Drake of England, Fire Over England, The Sea Hawk,* and *Das Herz der Königin.* If her father's ascension of the throne was improbable, hers was even more so. Henry had parliament legislatively bastardize Mary and Elizabeth when he divorced their mothers; however, he restored them by the Succession Act of 1543 and his will. Nevertheless, an air of uncertainty hung over both, given doubts about the legitimacy of Henry's first two marriages. There were no questions about Edward, who succeeded to the throne at age nine and saw England become genuinely Protestant, but died aged sixteen in 1553. His attempt to divert the succession to his Protestant cousin Jane Grey failed, and Mary restored Catholicism, married Philip II of Spain, burned some 300 heretics, lost Calais to the French, and experienced a false pregnancy before dying in 1558. It was unclear until the last minute whether Elizabeth would follow her on the throne, and even though she reigned for forty-five years, she nearly died of smallpox in 1562 and faced constant threats from disgruntled subjects, hostile foreign monarchs, Catholic recusants, Puritan dissenters, conspirators determined to replace her with Mary, Queen of Scots, and Imperial Spain. She also faced the challenge of being a woman in a male-dominated world, particularly her privy council and parliament, and had to resist pressure to marry and further reform the Church of England, make war against France or Spain, support continental Protestants to a greater degree than she was willing, execute her prisoner of nineteen years, Mary Stuart, Queen of Scots, before she was ready to do so, and to name a successor, which she refused until on her deathbed.[19]

The ardent Catholic Mary Stuart had a claim to the English throne through her father James V, son of Henry VII's daughter Margaret and James IV. She inherited the throne of Scotland six days after her birth in 1542, fled to France to escape the Rough Wooing, married the future Francis II in 1558, became Queen of France in 1559, and then a widow on his death in 1560. The French encouraged her to claim the English throne, arguing that Elizabeth was illegitimate and a heretic. Mary antagonized Elizabeth in 1565 by marrying her cousin Henry Stuart, Lord Darnley, who also could trace his line back to Henry VII through his mother Lady Margaret Douglas, daughter of Margaret Tudor by her second husband Archibald Douglas, 6th Earl of Angus. This meant their son, the future James VI and I, had a claim to the throne through both parental lines. Mary was suspected of involvement in the murder of her husband in 1567, married another suspect, James Hepburn, Earl of Bothwell, and provoked a rebellion that forced her to flee into England seeking safety in 1568. Instead, Elizabeth kept her prisoner for nineteen years, during which Mary was constantly involved in plots to overthrow Elizabeth and take the throne herself. Elizabeth finally had her executed in 1587.[20]

---

[18] The standard biography of Henry VIII is (Scarisbrick 1997); see also (Elton 1962; Coleman and Starkey 1986; Betteridge and Freeman 2016; Betteridge and Lipscomb 2013; Rankin et al. 2009; MacCulloch 2018).

[19] (Robison 2017).

[20] The most recent scholarly biography of Mary, Queen of Scots is (Guy 2004).

Thereafter, Elizabeth almost immediately faced the threat of invasion by the Spanish Armada in 1588. The victory of her navy, assisted by what the English interpreted as a God-sent storm, is perhaps England's most famous maritime triumph, and its anniversary was long one of the nation's most popular holidays. Though Spain sent additional armadas in 1596, 1597, and 1601, none was successful. Elizabeth ruled another fifteen years until 1603, facing increasing problems related to age and ill-health, court factions, shortage of funds, quarrels with Catholics, Puritans, and parliament, and rebellion in Ireland. However, she died peacefully in her bed. Many historians regard her as the greatest English monarch, and she contends with Churchill for the title of greatest person in English history. She was perfect as a subject for pro-British propaganda or as a target for the Germans.[21]

A key figure in defeating the Armada and another useful subject for propaganda was the explorer, pirate, and Elizabethan naval officer Drake. Born c.1540, eldest son of a shearman named Edmund Drake, he sailed on slaving ships with his kinsman John Hawkins between 1560 and 1568, spent several years raiding Spanish colonies in the New World, circumnavigated the world in 1577–1580 and thereby enriched the queen, was knighted and elected mayor of Plymouth in 1581, represented the town in parliament in 1581 and again in 1584, and after the death of his first wife in 1583, married Elizabeth Sydenham in 1584. He led an extended raid in the West Indies in 1585–1586, which in effect began an undeclared war with Spain. Elizabeth and Philip already were engaged in indirect hostilities in the Netherlands, where the queen provided support for the Dutch rebels, and in France, where England backed the Huguenots against the Catholic League. This escalated in 1587, when Drake "singed the king's beard" with a devastating raid on the port of Cadiz, delaying his plan for a maritime attack on England. However, Elizabeth's execution of Mary, Queen of Scots in 1587 gave the king a rationale for an invasion and an additional incentive, as he could now place one of his own family on the English throne. Fortunately, Drake had gathered enough intelligence to warn Elizabeth of Philip's plans. After helping defeat the Armada, he squandered his glory and infuriated the queen with his failed expedition to Portugal in 1589. However, after several years he made one last voyage to the Indies, on which Hawkins died in late 1595 and Drake himself in 1596.[22]

Drake, then, is yet another historical figure who can be cast as either hero or villain. As the former, he appears in his own right in *Drake of England*, but he is also clearly the inspiration for the characters Michael Ingolby in *Fire Over England* and Geoffrey Thorpe in *The Sea Hawk*. All deliver the message that England must be prepared to defend itself against a continental enemy, use Philip II as an analog for Hitler, and—less understandably—make William Cecil, Lord Burghley the advocate of appeasement. It might seem odd that Germany did not make use of Drake in its effort to demonstrate the historical greed and brutality of Britain, for he is as much a symbol of imperialism as of patriotism, and he began his sailing career on slave ships. However, it is only since World War II that historians have put as much emphasis on Drake the slaver as they have on Drake the adventurer and military hero.[23]

## 5. *The Private Life of Henry VIII* (1933)

Though the preceding historical survey proceeds chronologically, it seems best to deal with the seven films in order of release. Notably, both Britain and Germany produced films about the Lancastrians and the Hundred Years War and films set during Elizabeth's reign, but neither made one about the Yorkists and the Germans no film about Henry VIII. As with Drake, the latter seems like a lost opportunity, for Henry's faults are a propagandist's dream; however, there already was a feature-length German treatment of his reign, the 1920 silent film *Anna Boleyn*. Its director was Ernst Lubitsch, an Ashkenazi Jew who left Germany for Hollywood in 1922. It starred in the title role Henny Porten, who encountered Nazi hostility because she refused to divorce her Jewish husband, Wilhelm

---

21　Among many important new works on Elizabeth's reign are (Doran 2015; Guy 2016).
22　One of the best among many recent biographies of Drake is (Kelsey 1998).
23　For example, (Hazlewood 2004).

von Kaufmann. Her costar as Henry, Emil Jannings, later had a brief career in Hollywood before returning to Germany and appearing in pro-Nazi films. The reason there were no Yorkist films, despite Richard III being an ideal villain, is probably that during the Wars of the Roses England had no major foreign war suitable as an analogy for modern conflict.[24]

Korda's London Films released the first of the seven films, *The Private Life of Henry VIII*, on 21 September 1933, approximately eight months after Hitler became Chancellor of Germany on 30 January. By the time the film appeared, Hitler had announced his goal of obtaining *lebensraum* (living space) for the German master race, blamed the Reichstag Fire of 27 February 1933 on the Communist Party and banned it, secured the power to make law without the Reichstag for four years, created the Gestapo, installed Nazis in local government, banned trade unions and all other political parties, and withdrawn Germany from the League of Nations. Meanwhile, Goebbels encouraged mass burnings of 'un-German' books. Despite all this, Korda was one of the few who took seriously Churchill's warning about the existential threat the Nazis posed to Britain, where a multi-party National Government headed by Ramsay MacDonald favored appeasement.[25]

Korda used the film to emphasize this peril, though that might not be apparent to a twenty-first century viewer. Charles Laughton was already a well-established actor when Korda cast him as Henry. He did considerable research on the king, though some elements of his characterization now seem off the mark. Korda cast Laughton's wife Elsa Lanchester as Anne of Cleves, who usually gets short shrift in films, but features prominently in this one. Seeking to make a star of Merle Oberon, his future second wife, he cast her as Anne Boleyn, who usually has a larger role. Also cast were Wendy Barrie as Jane Seymour, Binnie Barnes as Catherine Howard, and Everley Gregg as Catherine Parr. The film now seems quaint and almost innocent in its depiction of Henry's reign, often resembling high society films of the 1930s. It mingles elements of comedy, pathos, and sexual titillation with analogies to modern British foreign policy and decidedly noncomic events, as when a husband and wife exchange humorous remarks while in the audience for Anne Boleyn's execution. It presumes the nascent British Empire is a good thing without a hint of the anti-imperialism that in recent years has characterized post-modern and post-colonial studies. It contains misogynistic elements, for example, Henry tells Culpeper, "Thomas, if you want to be happy, marry a girl like my sweet little Jane. Marry a stupid woman." However, it understates Henry's grief over Jane's death. A success at the box office and among critics, it got an Academy Award nomination for Best Picture, won a Best Actor Oscar for Laughton, made stars of Korda and Laughton, was good anti-German propaganda, and remains watchable today.[26]

*Private Life* reflects three interlinked influences that are examples of Tudorism: "[t]he post-Tudor mobilization of any and all representations, images, associations, artefacts, spaces, and cultural scripts that either have or are supposed to have their roots in the Tudor era." First, sixteenth-century artist Hans Holbein the Younger created a popular image of Henry surpassing all others—the massive, confident-looking fellow with the broad shoulders, wide stance, elaborate regalia, assertive codpiece, and hand resting on his dagger. Second, stage productions—especially of William Shakespeare, *The Famous History of the Life of King Henry the Eight*—created a behavioral stereotype to match, incorporating the Holbein image with well-documented, though frequently exaggerated, elements of Bluff King Hal's personality to produce a larger-than-life, Falstaff-like character with gargantuan appetites and outsized emotions, whether joy or sorrow, love or hatred, satisfaction or anger. Third, Laughton drew on early performances, especially Jannings, as well as adding his own touches. Holbein's portrait and Shakespeare's play are also propagandistic, which makes for a lot of propaganda. Because

---

[24] See especially (Walker 2003, p. 12); also (Parrill and Robison 2013, p. 14).
[25] See especially (Walker 2003, chp. 1); also (Chapman 2005, chp. 1; Freeman 2008; Harper 1994, chp. 2; Parrill and Robison 2013, pp. 180–81).
[26] Korda's frequent collaborators Lajos Biró and Arthur Wemperis wrote the screenplay, though Francis Hackett later claimed they had plagiarized (Hackett 1929); see also (Walker 2003, chp. 2).

virtually every subsequent filmic Henry has been part Holbein, part Shakespeare, and part Laughton, it is easy for viewers to accept that artificial and constricted depiction of a very complex man.[27]

Most films about Henry focus on the love triangle involving the king, his first wife Catherine of Aragon, and his mistress and later second wife Anne Boleyn. *Private Life* is different, ignoring Catherine, giving short shrift to Anne Boleyn, and concentrating on the last four wives, especially the German Anne of Cleves. The action ensues with Anne Boleyn already in the Tower, awaiting execution by a bizarrely cheerful French swordsman, whom the usual English executioner berates as a foreigner. Korda intercuts scenes of Anne readying herself for the block, Jane preparing for her wedding, and Henry discoursing to Culpeper on women and to the council on England's need for more ships: "No, Cromwell, if England were as rich as Portugal or as big as Spain you might be right, but this little island of three million souls is no match for Europe. If these French and Germans stop cutting each other's throats, what's to stop 'em cutting ours?" Cromwell replies, "Wise diplomacy, Sire." Stamping his foot, the king rejoins, "Diplomacy? Diplomacy me foot! I'm an Englishman; I can't say one thing and mean another. But what I can do is to build ships, ships, and more ships." Cromwell asks, "You mean double the fleet?" Henry, in full swing, ups the ante: "Treble it! Fortify Dover. Rule the sea!" A worried looking Cromwell observes, "To do this will cost us money, Sire." Henry then proclaims, "To leave it undone will cost us England." His councilors nod approvingly. No appeasement here.

In his excellent film guide for *Private Life*, Greg Walker shows that Korda's exercise in peacetime propaganda served multiple, related purposes. It responded to the 1932 call from Sir Stephen Tallent, future director general of the Ministry of Information, for the 'projection of England' or traditional 'Englishness' against the corrosive effects of American culture, expressed sympathy with unemployed Englishmen during the Great Depression, and presented Henry as a forthright and patriotic national statesman keenly aware of England's danger and at least by implication wiser than those who supported appeasement of Hitler and his allies. Indeed, as Walker notes, it intervened in ongoing parliamentary debates about foreign policy, in which the appeasers enjoyed far more support among members than Churchill with his dire forecasts about the need for rearmament in the face of the Nazi threat to peace. It also pokes fun at Germans.[28]

The plot moves quickly from Henry's marriage to Jane to the birth of Prince Edward and his mother's death. Then comes the famous banquet scene, in which Henry sloppily devours roast capon while lamenting the decline of manners and denouncing those who want him to marry a fourth time and produce more sons. But soon his ministers persuade the king, who usually married for love, to seek a bride for the sake of diplomacy. He sends Hans Holbein the Younger to paint a portrait of his prospective German bride, Anne, sister of the Duke of Cleves. The film portrays the duke as a gangly fop in a silly outfit who speaks broken English and unwittingly reveals that Anne may have lost her virginity to her fictional lover, Thomas Peynell. Anne is more interested in Peynell than Henry, takes him with her to England, and plots to make herself look ugly so the king will be repulsed. There is more humor at German expense when she and her entourage reach England. Henry finds her unattractive ladies in a sort of receiving line, wrongly assumes that one of them is Anne, and is horrified. She comes in disheveled, face distorted, pretending seasickness, and kisses Henry, who recoils and dismisses her, after which he rails at Cromwell.

The wedding night is a comic masterpiece in which Anne plays ignorant about sex, Henry awkwardly tries to explain, the two play cards, and he loses, but she agrees to a divorce, and they part on good terms. During the scene Anne tells Henry she knows that he is pursuing Catherine Howard, who—contrary to history—has been around since the start of the movie, is older than in reality, is ambitious to be queen, and has flirted with Henry, who responds like a school boy. All of this is funny but has little to do with propaganda. Eventually, as in reality, Catherine is caught in

---

[27] See especially (Walker 2003, pp. 18–26); the Folger Shakespeare Library version of the play is (Shakespeare 2007; String and Bull 2011, p. 1; String 2011, 2013).
[28] See (Walker 2003, chp. 3).

adultery with Culpeper. Sad and old, Henry chats with Anne of Cleves, who recommends he pay court to Catherine Parr, who at least in this movie proves to be a shrew who nags the kings about his health, or as Henry puts it, "Six wives and the best of them is the worst."

## 6. *Drake of England* (1935)

On 16 May 1935, less than a month before MacDonald yielded the prime ministry to Stanley Baldwin, Elstree Studios released *Drake of England* (a.k.a. *Drake the Pirate*). By this point, Hitler had purged the Nazi Party of Ernst Röhm and the *Sturmabteilung* (SA), combined offices of Chancellor and President in his person as Führer, begun rearming, had Hermann Goering begin developing the Luftwaffe, and introduced conscription. The film did not involve Churchill, Korda, or Olivier but has much in common with *Fire Over England* and *The Sea Hawk.* Arthur B. Woods directs, Matheson Lang plays Drake much like Errol Flynn, Athene Saylor portrays the queen, and Jane Baxter is Elizabeth Sydenham. The film anachronistically incorporates Edward Elgar's *Pomp and Circumstance* and intermingles scenes of Drake's attacks on Spanish interests, courtship of Elizabeth Sydenham, and advocacy of a stronger naval force against 'appeasers' like Burghley.[29]

It opens in 1567, with Spain mistress of the seas and Drake and Hawkins—whose slaving is not mentioned—"determined to challenge the might of Spain". Elizabeth Sydenham visits Plymouth harbor to see Drake's ship, and a stereotypical matron named Mother Moone (Amy Veness) takes her to task for entering this male realm. However, Elizabeth clearly is a rebel and smitten with Drake. John Doughty (Henry Mollison) conspires with the Spanish envoy Don Bernardino de Mendoza (Allan Jeayes) to put Mary, Queen of Scots on the throne. Drake arrives, fresh from conflict with Spain at San Juan de Ulua, and when the queen asks why, he responds, "Rage, madam, and hunger for bloody vengeance." Action shifts to Drake's 1572 expedition to Nombre de Dios, followed by his return to Plymouth and a 1573 appearance at Windsor Castle. There is much ado with Elizabeth Sydenham, whom Drake marries in 1577 (in reality 1585) before embarking on his circumnavigation, which the queen conceals from Burghley (Ben Webster). The voyage features a near mutiny and Drake's execution of Thomas Doughty (Donald Wolfit), but all ends well with his return to Plymouth and a visit from the queen. She knights Drake, saying, "Thus do I honor the man who opened the seas of the world to English ships and taught Englishmen to be sailors." She also approves Drake's marriage, and Mendoza leaves in a huff.

Subsequent scenes show the Spanish building ships at Cadiz, and Mother Moone mentions Drake singeing the king's beard, but the raid does not take place onscreen. However, viewers do see Elizabeth receiving intelligence confirming the Spanish threat, and she orders the Lord Admiral (Ian Fleming) to seek Drake's advice. The film shows Drake's famous game of bowls, which legend has him completing before going off to fight the Armada. The battle is shown at some length, both off Plymouth and at Calais, though with some inaccuracies and anachronisms. The scene then shifts to Tilbury, where Elizabeth gives a somewhat altered version of her famous speech. Drake arrives and declares before the queen, "The Invincible Armada is vanquished. We have opened the gates of the sea and given you the keys of the world. From this day forward the English merchant can rove whither he will and no man shall say him nay. The little spot ye stand on has become the center of the earth. Men of England, hitherto we have been too much afraid. Henceforth we will fear only God." Everyone cheers, and the film ends. The message could not be more obvious.

## 7. *Das Mädchen Johanna* (1935)

*Das Mädchen Johanna* appeared on 8 October 1935. In the interval Stanley Baldwin replaced MacDonald as prime minister, while Hitler duplicitously accepted the Anglo-German Naval Agreement and instituted the Nuremberg Laws banning German-Jewish relationships. Director Gustav Ucicky had

---

[29]  See (Doran 2008, pp. 91–92; Latham 2011, p. 93; Parrill and Robison 2013, pp. 42–44).

displayed intense nationalism on film before Hitler took power and went on to make Nazi propaganda movies from 1933 through 1945. Screenwriter Gerhard Menzel contributed to pro-Nazi films as well. Angela Salloker, who played Joan, was an Austrian actress who later appeared with Emil Jannings in Ucicky's production of *Der Zerbrochene Krug* (*The Broken Jug*, 1937), a historical comedy and one of Hitler's favorites. Gustaf Gründgens costarred as Charles VII. Goebbels had given him the honorary title of *Staatsschauspieler* (State Actor), which he soon bestowed on other actors in the film, including Bernhard Minetti (Amtmann), Erich Ponto (Lord Talbot), Heinrich George (Herzog von Burgund, or the Duke Burgundy), Theodor Loos (Dunois), and Willy Birgel (La Trémouille).[30]

It is a peculiar and highly inaccurate film. It begins at the camp of Lord Talbot, i.e., Sir John Talbot, 1st Earl of Shrewsbury, who in reality was a fierce warrior, honored by the English, feared by the French, and praised in Shakespeare's *Henry VI, Part 1* as "valiant Talbot" and "terror of the French". In the film, however, he is clownish, cowardly, and incompetent. In the first scene he quarrels with Burgund (who bears a striking resemblance to the bumbling Sergeant Schultz played by John Banner in the CBS television comedy *Hogan's Heroes*) over who should have the spoils of Orléans, quickly establishing that both are greedy dolts. Burgund hears a noise, thrusts his sword through the tent wall, and kills Talbot's man Billy, after which they negotiate over the value of his life. The fictional Maillezais (René Deltgen), whom Talbot denounces as the son of a laundress, delivers a message from Charles. However, Talbot sends him back empty-handed after branding him with a hot coal on the forehead.

The scene shifts to Orleans, where Maillezais tries to convince a group of the king's advisors—Valençon (Jean II, Duke of Alençon), La Trémoille (George de la Trémoille, Count of Guînes and Great Chamberlain to Charles VII), and Dunois (Jean d'Orléans, Count of Dunois, 'the Bastard of Orléans)—that an English attack is coming. The king arrives, and his advisors greet him with derision. He and Maillezais depart in Valençon's palanquin, and a mob mistakes him for the duke and attempts to kill him. Just in time church bells begin to ring, the rioters drop to their knees, and Joan appears accompanied by Johann de Metz. In effect, she rescues the king and then tells him that the Archangel Michael has sent her to help him take the throne. There is no mention of Chinon, where they actually met, and no appearance by Robert Baudricourt, who brought her to meet Charles, or the king's mother-in-law Yolande of Aragon. Joan leads the people and Charles' soldiers in routing the English, at one point climbing a scaling ladder up the wall of a fortress as its defenders pour boiling pitch over the side.

Later, as Talbot flees in terror, Charles arrives at his tent, where Burgund begs for mercy and offers an alliance with France that did not actually occur until 1435. The king presents Joan with a silver suit of armor, but without any explanation the crowd turns against her. At a banquet she sits with the king and La Trémoille while the roaring drunk Burgund and Valençon argue and insult each other amid great hilarity. La Trémoille learns that the Black Plague has erupted in town, and everyone blames Joan, accusing her of being a witch. She attempts to escape with Maillezais, but is caught. Charles repudiates her but later denounces and imprisons La Trémoille. Meanwhile, Dunois fails in his attempt to rescue Joan. After an initial bout of anguish, she bravely faces the fire, refusing to deny she has been guided by the Archangel Michael. The film ends in 1456 with Charles announcing that the Church has overturned the charges against Joan.

The film is not remotely subtle in presenting the English and French as self-serving idiots. While it is by no means an allegory exactly paralleling Joan and Hitler, both the German press and foreign media noted the similarities between the heroism of her 'liberation' of France in 1429 and Hitler's reassertion of German sovereignty in 1933–1935. Interestingly, Joan, who dressed like a man and was charged with cross-dressing among her other errors, has dark hair cut like the Führer's. In *The Spectator*, Graham Greene criticized the film's inaccuracies, compared Gründgens' characterization of Charles to Hitler's cunning but cruel behavior, and excoriated what he perceived as analogies between the fall of

---

30　See (Fox 2000, p. 24; Rosenstone 2003).

La Trémoille and the Night of the Long Knives and between Joan's burning and the Reichstag Fire. While the movie has numerous problems, it is at times rather funny and was popular in Germany, particularly among young people.[31]

### 8. *Fire Over England* (1937)

On 5 March 1937 London Films released *Fire Over England*, a film similar to *Drake of England* but with better production values. Meanwhile, in 1936 Germany put the Gestapo above the law and occupied the Rhineland in violation of the Treaty of Versailles, Mussolini took Ethiopia, Civil War broke out in Spain, and the famed 1936 Olympics took place in Berlin. British appeasement continued, with Neville Chamberlain waiting in the wings to replace Baldwin. Korda and Erich Pommer produced the film, American filmmaker William K. Howard directed, and Clemence Dane and Sergei Nolbandov wrote the screenplay, loosely based on A.E.W. Mason's novel. Flora Robson, cast as Elizabeth I, had previously appeared as the Empress Elizabeth in *The Rise of Catherine the Great* (1934), which Korda produced, London Films released, and Nazi Germany banned because Catherine Bergner, who portrayed Catherine, was Jewish. Raymond Massey is Philip II. Fictional characters are more numerous, including Olivier as Michael Ingolby, Vivien Leigh as Cynthia, and Tamara Desni as Elena. While Ingolby's character is clearly based on Drake, there are references to Drake himself in the film.[32]

The message Korda intended to deliver is more pronounced than with *Drake of England* or *Das Mädchen Johanna*. A Spanish ship commanded by Don Miguel (Robert Rendell) captures a ship commanded by his friend Sir Richard Ingolby (Lyn Harding) and allows his son Michael Ingolby to escape. He ends up on an estate belonging to Don Miguel, whose daughter Elena nurses him back to health. The two are smitten with each other even though Elena is engaged to Don Pedro (Robert Newton) and Michael misses his sweetheart Cynthia. Michael's father sends a message from captivity telling him, "It isn't our quarrel. It isn't your quarrel. It is a war of ideas. You can't burn ideas." But, unfortunately, the Spanish Inquisition does burn Sir Richard. Eventually Michael escapes to England and Cynthia, the only lady-in-waiting visible in the film. Romance takes up a good bit of screen time, and there is an obvious contrast between young Michael and Cynthia and the aging queen who self-consciously bans mirrors from her presence and the Earl of Leicester (Leslie Banks). More pertinent to this essay, however, is that Elizabeth, who initially complains that Michael smells like fish, listens to his pleas that she prepare to fight against Spain, as opposed to the concerns of Lord Burleigh (Morton Selten) about the cost of expanding the fleet. When a plot to kill her is uncovered, she sends him back to Spain to recover letters belonging to a dead conspirator, Harry Vane (James Mason), and identify his associates.

Disguised as Vane, he goes to Philip II's court, where he has close encounters with the king and his fanatical minions. He meets Elena, who conceals his identity for a time but eventually reveals it to her husband, the palace governor. However, it is Philip who realizes that Michael is a spy and has him arrested, while Don Pedro helps him escape again so that the king will not learn that Elena previously assisted a heretic. Michael finds Elizabeth at Tilbury, where as usual she gives her famous speech. He gives her the names of the traitors, and—after knighting him—she confronts them, they beg forgiveness, and she gives them the chance to accompany Michael on his mission to send fireships against the Armada at Calais. This succeeds, Michael survives, and the queen allows him to marry Cynthia.

The film obviously alludes to the possibility of a Nazi invasion like Hitler planned with Operation Sea Lion but aborted after British success in the Battle of Britain in 1940. Korda later included portions

---

31　See (Fox 2000, p. 24; Taylor 1980).

32　See (Betteridge 2003; Dobson and Watson 2002, pp. 209–12; Doran 2008; Coster 2008; Ford 2009, pp. 229, 235; Latham 2011, pp. 65–76; Parrill and Robison 2013, pp. 86–87; Vidal 1992, pp. 31–64); for further discussion of Philip II on film and in the Black Legend, see (Robison 2016b).

of the remarkably prescient film in the 1939 propaganda documentary, *The Lion Has Wings*, which recounted past victories to encourage the British people to optimism. The film also leaves no doubt about Philip, who clearly represents Hitler. Viewers learn that the Spanish king "rules by fear" and that "Spanish tyranny is challenged only by the free people of a little island". Fire imagery appears throughout, with Ingolby warning Elizabeth that if she dies, there will "fire over England". He hears Philip comment, "Only by fear can the people be made to do their duty, and not always then," and Elena's husband observes, "The whole trouble comes from treating your enemies like human beings. Don't you see, my dear, that if you do that they cease to be enemies. Think what that leads to: the end of patriotism; the end of war; it's the end of everything." At one point the Spanish ambassador tells Elizabeth, "If your majesty will not hear words, we must come to cannon and see if you will hear them," and is taken aback when she threatens to drive him out of England.

## 9. *The Sea Hawk* (1940)

*Drake of England* was released in the United States in 1936 and *Fire Over England* in 1937. In 1940 Warner Brothers released a very similar American film, *The Sea Hawk*, with the intent of encouraging the British people and increasing American support for Britain in the war. In the three years between 1937 and 1940 much had happened. In 1937 Joseph Stalin conducted a purge of the Red Army, Hitler revealed his plans for war to select German commanders at the secret Hossbach Conference, and Japan invaded China, beginning the Second Sino-Japanese War (1937–1945) and carrying out the mass atrocity known as the Rape of Nanjing. In 1938 Germany completed the Anschluss whereby Austria united with Germany, British Prime Minister Neville Chamberlain agreed at the Munich Conference to Hitler taking the Sudetenland in Czechoslovakia, the German army occupied it, German and Italian 'volunteers' fought in Spain, and there was a mass attack on German Jews known as Kristallnacht on 9 November. In 1939 Hitler openly threatened Jews in a Reichstag speech, Nazi forces seized the rest of Czechoslovakia, the Spanish Civil War ended with Franco firmly in power, Hitler and Mussolini signed the Pact of Steel, Germany and the Soviet Union signed the Molotov-Ribbentrop Pact, Britain signed a mutual assistance treaty with Poland, Germany invaded Poland from the west on September 1 and the Soviet Union invaded from the east on September 17.[33]

By the time *The Sea Hawk* appeared in theaters on 31 August 1940, Britain was the only European state left to oppose Hitler, its allies being limited to Canada, Australia, and New Zealand. Churchill had been prime minister for just three-and-a-half months. Rationing had begun. Germany and the Soviet Union were still bound to each other by the Molotov-Ribbentrop Pact, for another nine months as it turned out. The Germans had overrun Poland, Denmark, Norway, the Netherlands, Belgium, Luxembourg, and France. U-boats were attacking merchant shipping in the Atlantic Ocean, and Hitler had declared a blockade of the British Isles. The Battle of Britain was in full swing, with the Luftwaffe bombing British airfields and factories and then conducting night raids on London. The Soviet Union had seized a portion of Finland and occupied Latvia, Lithuania, and Estonia. Mussolini had entered the war as Hitler's ally, and his troops had taken British Somaliland. America's entry into the war was still over fifteen months away, and President Franklin Roosevelt was in the midst of a reelection campaign against Republican Wendell Wilkie in which a major issue was noninterventionism promoted by the America First Committee with the support of Charles Lindbergh. It was a grim moment.

In *The Sea Hawk*, Robson reprised her role as Queen Elizabeth, with Michael Curtiz directing his longtime collaborator Errol Flynn as Geoffrey Thorpe, another stand-in for Drake in another swashbuckling maritime film set at the time of the Armada, Barbara Marshall as Doña Maria, and Montagu Love as Philip, who again represents Hitler. As with *Fire Over England*, many of the characters are fictional. As the movie begins, Philip is planning to conquer England and then the

---

33 See (Dobson and Watson 2002, pp. 279–80; Doran 2008, pp. 94–97; Coster 2008; Ford 2009, pp. 229, 235; Latham 2011, pp. 90–101; Parrill and Robison 2013, pp. 197–99).

world, but he sends the fictional Don José Alvarez de Cordoba (Claude Rains) to England to convince Elizabeth of the opposite. Some of her subjects urge her to build a fleet that can resist the Armada Philip is constructing, but she hesitates. Thorpe, aboard the *Albatross,* captures the ship carrying Cordoba and his niece Doña Maria, who gradually falls in love with him. Cordoba complains about English 'sea hawks' plundering Spanish shipping, Doña Maria becomes one of the queen's ladies, and Thorpe proposes a plan to seize Spanish gold coming from the New World.

Lord Wolfingham (Henry Daniell), a fictional royal minister but also a spy for Spain, sends a ship ahead to Panama, where the Spanish ambush Thorpe. They are sent to Spain, where the Inquisition sentences them to serve for life as galley slaves for their heresy. Thorpe and another Englishman named Abbott (James Stephenson) seize control of a galley, capture a passenger who is an envoy, and seize his papers, which reveal a plot against Elizabeth. When he reaches England, Cordoba comes unawares to meet the Spanish ship, and Thorpe captures him again. Doña Maria helps him to slip into the palace, where he fights with Wolfingham's men with a great deal of the swashbuckling fervor that made Flynn famous. Eventually he fights and kills the evil councilor Wolfingham. After Thorpe shows Elizabeth evidence of Philip's intentions, she promises to build a fleet to combat the Armada.

In the version shown in England, Robson gives a long patriotic speech at the end. It is worth quoting at some length.

"And now my loyal subjects, a grave duty confronts us all: To prepare our nation for a war that none of us wants, least of all your queen. We have tried by all means in our power to avert this war. We have no quarrel with the people of Spain or any other country; but when the ruthless ambition of a man threatens to engulf the world, it becomes the solemn obligation of all free men to affirm that the earth belong not to any one man, but to all men, and that freedom is the deed and title to the soil on which we exist. Firm in this faith, we shall now make ready to meet the great that Philip sends against us. To this end, I pledge you ships—ships worthy of our seamen—a might fleet hewn out of the forests of England; a navy foremost in the world—not only for our time, but for generations to come."

These are words worthy of the wartime Churchill at his finest.

## 10. *Das Herz der Königin* (1940)

Ufa released *Das Herz der Königin* two months later on 1 November, by which time Hitler had launched and then postponed Operation Sea Lion (a plan for invading Britain), Germany signed the Tripartite or Axis Pact with Italy and Japan, German troops invaded Romania, and the Italians invaded Egypt and Greece. There were additional releases in occupied states over the next year. Director and producer Carl Froelich was a well-established filmmaker long before the Nazis came to power. His first film, appropriately enough, was about the composer, German nationalist, and anti-Semite Richard Wagner, whose music Hitler loved. He made the first German sound film, *Die Nacht gehört uns* (*The Night Belongs to Us*, 1929) and the first color drama, *Das Schönheitsfleckchen* (*The Beauty Spot*, 1936). In 1933 he joined the Nazi Party, took over direction of the *Gesamtverbandes der Filmherstellung und Filmverwertung* (Union of Film Manufacture and Film Evaluation), became a professor at the *Reichsfilmkammer* in 1937 and its president in 1939, and was honored by Goebbels. He made several historical films, including *Luise, Königin von Preußen* (*Louise, Queen of Prussia*, 1931), *Der Choral von Leuthen* (*The Hymn of Leuthen*, 1933), *Liselotte von der Pfalz* (*Liselotte of the Palatinate or The Private Life of Louis XIV*, 1935), *Heimat* (1938), *Es war eine rauschende Ballnacht* (*It Was a Lovely Night at the Ball or The Life and Loves of Tschaikovsky*, 1939).[34]

There was a long history of theatrical productions about the rivalry between Elizabeth I and Mary, Queen of Scots. Friedrich Schiller's play *Maria Stuart* (1800), which inspired Gaetano Donizetti's opera

---

34 The film was released in Hungary on 19 December, the Netherlands on 20 December, Denmark on 10 January 1941, Sweden on 30 January, Finland on 16 February, and France on 13 November, and actually was shown in the United States in 1948, https://www.imdb.com/title/tt0032587/; see also (Bock 2009, pp. 137–38; Fox 2000, pp. 25–31; Parrill and Robison 2013, p. 107).

*Maria Stuarda* (1835) and the silent short *Mary Stuart* (1913), was pro-Mary, as was the first sound film, John Ford's *Mary of Scotland* (1936), which portrays Mary (Katherine Hepburn) as admirable and Elizabeth (Florence Eldridge) as Machiavellian. The screenplay for *Das Herz der Königin*, based on a novel by Harald Braun, does likewise while making a thorough hash of history. The costumes by Herbert Ploberger range from plausible-if-inaccurate to bizarre in the case of English soldiers, and Theo Mackeben's score is bombastic, though hardly more so than *Deutschland über Alles*, and contributes to the film's pro-German mood.[35]

Zarah Leander, who plays Mary, was a Swedish singer and actress who became one of the biggest stars in Germany during the Nazi era. After making several films in Sweden, she signed a contract with Ufa in 1936. Her willingness to work for Goebbels allowed her to fill the niche that Marlene Dietrich vacated when she abandoned Germany for Hollywood in 1936 and that her Swedish contemporaries Greta Garbo and Ingrid Bergman eschewed. Prior to *Das Herz der Königin* she worked with Froelich on *Heimat* and *Es war eine rauschende Ballnacht* and Willy Birgel, who played Lord Bothwell, on *Zu neuen ufern* (*To New Shores*, 1937) and *Der Blaufuchs* (*The Blue Fox*, 1938). Mary was a thin eighteen-year-old when she returned from France to Scotland in 1561, whereas Leander was a heavy-set thirty-three and looked older in the film. She also insisted upon singing, so there are several sad, rather banal songs with no cinematic rationale. Her husky, Dietrich-like voice is always slightly flat, so she sounds very much like Madeline Kahn's Lili von Shtüpp the 'Teutonic Titwillow' in Mel Brooks' *Blazing Saddles* (1974) and is unintentionally comical to the present-day viewer.[36]

Maria Koppenhöffer is appropriately cold and self-contained as Elizabeth, whereas Willy Birgel is melodramatically sinister as Bothwell. Lotte Koch, who plays his wife Johanna Gordon, is dressed in men's clothes and wears a sword for most of the film, calling to mind that Froelich was art director for *Mädchen in Uniform* (1931), which the Nazis later banned for 'decadent' lesbian content. Perhaps following the example of Douglas Walton in *Mary of Scotland*, Axel von Ambesser portrays Darnley as effeminate and perhaps gay, a trend that has continued in subsequent films down to *Mary, Queen of Scots* in 2018. Friedrich Benfer as David Riccio does not appear Italian, and Ernst Stahl-Nachbaur is disappointingly low-key as Protestant firebrand John Knox, though Walther Süsseguth is acceptable as Mary's half-brother Lord Jacob (James Stuart, 1st Earl of Moray, regent from the death of her mother Marie of Guise in 1560 until Mary's return).

The film opens with Mary in prison, mournfully singing to her attendants. She is tried by the fictional 'Supreme Crown Court' of England, made up of dour old men for whom a guilty verdict is a mere formality. Elizabeth is not present during the trial, but the judges pay reverence to a cushioned stool holding two closed imperial crowns and a short sword. Mary refuses to believe that Elizabeth will have her executed, swearing repeatedly that she is not England's enemy and is innocent of the charges against her. The action then flashes back to her arrival in Scotland, where the people regard her skeptically, Lord Jacob advises her to return to France, Knox condemns her Catholicism, the lords refuse to cooperate, Johanna attempts to poison her, and Bothwell tries to seduce her. Though she haughtily rejects him, for reasons never explained she subsequently is attracted to him. When the lords continue to thwart her, she turns the tables by promising them a king, with the intent of marrying Bothwell. Meanwhile, however, he has married Johanna. She turns in desperation to Darnley, whom the film portrays as a witting agent of Elizabeth, who has sent him to engineer Mary's downfall. There is no indication that their relationship is romantic, but suddenly they have a son, James. Viewers see nothing of Darnley's drunkenness, though he manages to be enraged just the same, first at street players lampooning his relationship with the queen and later at Mary for shutting him out of government. At times other lords refer to him as 'King Henry', but Mary pointedly tells him that he is only the

---

[35]　See (Doran 2008, pp. 97–98; Ford 2009, chp. 4; Latham 2011, pp. 41–65; Parrill and Robison 2013, pp. 144–45, 148).
[36]　See (Ascheid 2003, chp. 4; Bruns 2009, chp. 3; Carter 2007, chps. 6–7; Romani 2001, pp. 72–83).

queen's husband. He also becomes furious at Riccio, who—contrary to history and other films—is actually in love with Mary.

There is a bizarre scene in a bathhouse where several of the Scottish lords are being serviced in various ways by prostitutes, including baths, massage, and (off-camera) sex, possibly an oblique reference to the alleged decadence of the SA. Elizabeth's fictional messenger Sir John enters, refuses their invitations to join in the debauchery, but leads them to believe that Mary is sleeping with Riccio. Next the already angrily jealous Darnley arrives, and the lords convince him of the same thing. They murder Riccio, a horror from which Mary recovers rather quickly. She abandons Darnley for Bothwell, who likewise forsakes Johanna, and the lords turn against her. The calculating Elizabeth sends money to Lord Jacob to assemble an army and a message to Mary offering sanctuary in England to her and her son. Darnley contracts a venereal disease, Mary visits him seeking his support, and subsequently his house explodes in a rather unimpressive display of special effects. Lord Jacob seizes power, and Johanna gives him a casket full of letters between Mary and Bothwell, which he sends to Elizabeth. His troops then drag Bothwell to his death behind a horse, though in reality he fled to Denmark.

Without showing Mary's escape, the action shifts back to Mary in prison, where she makes a tearful farewell from her ladies and is led to the block. The film does not show her scarlet undergarments, which is surprising since it emphasizes her martyrdom, nor does it depict her execution or Elizabeth's subsequent remorse. Although the film offers no overt comparison to Hitler, the message is clearly there. Bothwell tells Mary that women are unfit to rule and that Scotland needs a strong rider. Her ladies observe that little Prince James will have strong fists, and Mary urges him to cry louder as it will make him strong.

## 11. *Henry V* (1944)

Four more years passed before the release of Olivier's *Henry V*. In that time Roosevelt was reelected twice, Japan brought America into the war by bombing Pearl Harbor, the Nazis invaded the Soviet Union and were repulsed, the Allies took control of North Africa and Italy, Chinese Communists and Nationalists combined forces against Japan, Allied forces advanced in the Pacific and bombed the Japanese mainland, and the Big Three of Churchill, Roosevelt, and Joseph Stalin agreed that Britain and America should launch a second front in Normandy. Though Churchill and Olivier intended that *Henry V* be in theaters in time for D-Day, which turned out to be on 6 June 1944, it was not released until 22 November. Still, its purpose of boosting British morale remained timely even if the Allies were now on the offensive.[37]

Both Olivier's *Henry V* and Shakespeare's *The Cronicle History of Henry the fift* (c.1599) are problematic in terms of historicity. The play is propagandistic in its own right, a good bit of the action involves fictional characters, and the bard's sources for historical persons and events were flawed chronicles by Edward Hall and Raphael Holinshed, the anonymous play *The Famous Victories of Henry the Fifth* (c.1594), and possibly Samuel Daniel's historical poem about the Wars of the Roses, *The First Four Books of the Civil Wars* (1595). It offers contradictory visions of the king as at times heroic and at others brutal and self-serving, which Norman Rabkin has symbolized with his double image of the rabbit and the duck. On top of that, the play follows *Henry IV*, Parts 1 and 2, in which Prince Hal, the future king, is a dissolute young man surrounded by Sir John Falstaff and other ne'er-do-wells, and hardly the stuff of which heroes are made. Olivier and co-writers cut the play to emphasize Henry's heroic nature and downplay his faults.

The film begins and ends with actors onstage c.1600 at the Globe Theatre, using the original title *The Chronicle History of King Henry the Fift with His Battell Fought at Agincourt in France*, with a

---

[37] On Henry onstage and onscreen (Buchanan 2005, pp. 189–200; Cartmell 2000, chp. 5; Chapman 2005, chp. 5; Davies 1990, chp. 2; Hindle 2007, pp. 30–31, 140–44; Jackson 2007, chp. 2; Puckett 2017, chp. 2; Rabkin 1977; Robison 2018).

dedication to "the Commandos and Airborne Troops of Great Britain the spirit of whose ancestors it has been humbly attempted to recapture". In Act 1: Prologue the Chorus (a single individual) calls for "a muse of fire" and urges the audience to use their imagination to move beyond the confines of the theater. In 1:1 a dignified Archbishop of Canterbury and the Bishop of Ely somberly discuss law, taxation, Henry's improved character, the French throne, and factional strife at court; however, on film the bishops are comical. In 1:2 Henry asks if it is legal to invade, the bishops and the Dukes of Essex and Westmorland encourage him, and after French ambassadors insult him with a mocking 'gift' of tennis balls from the Dauphin, he is ready.[38]

In Act 2: Prologue the Chorus praises English patriotism, in 2:1 Henry's old friends Bardolph, Nym, and Pistol visit the ailing Falstaff, who dies in 2:3. Olivier cut 2:2, in which Henry denounces the treason of the Earl of Cambridge, Baron Scrope, and Sir Thomas Gray, ignores their pleas for mercy, and has them executed. In Act 3: Prologue the Chorus describes the voyage to Normandy and notes that Henry has refused the French king's offer of his daughter's hand. In 3:1 Henry urges his troops on at Harfleur with the rousing lines, "Once more into the breach, dear friends, once more," and they respond with "God for Harry, England, and St. George". In 3.2 the Welsh Captain Fluellen forces Bardolph, Nym, and Pistol to fight, and Fluellen, Gower, the Irish MacMorris, and the Scot Jamy discuss mining. The film omits 3:3, in which Henry threatens murder, rape, and pillage in Harfleur unless it surrenders. In 3.4 Katherine tries to learn English from her maid Alice with hilarious results, and in 3.5 the French king and his courtiers alternately admire and insult the English troops. Olivier cut the portion of 3:6 in which Henry hangs his old friend Bardolph for theft. Henry then puts his fate in God's hands, and the army marches away.

In Act 4: Prologue the Chorus explains how Henry seeks to give his men courage, in 4.1 the king wanders among his soldiers in disguise, in 4.2 the arrogant French prepare for battle, and in 4.3 Henry delivers the rousing St. Crispin's Day speech in which he refers to his troops with the line, "We few, we proud, we band of brothers." Olivier stages this for maximum impact in the film. Henry also refuses the French herald Montjoy's offer of surrender terms. The battle does not occur onstage in the play, but does in the movie. In 4.4 the boy criticizes Pistol for ransoming a French captive, and in 4.5 the French are shamed by English success. Olivier cut 4.6 in which the English kill their French prisoners. In 4.7 the English discover that the French have broken the 'law of arms' by killing all the page boys, and Henry vows revenge, but Montjoy arrives to concede the English victory, and Henry plays a joke on Fluellen and Williams that comes to fruition in 4.8. In Act 5: Prologue the Chorus describes Henry's triumphant return to England and then France, in 5:1 Fluellen and Pistol fight, and in 5:2 the English and French discuss peace, and Henry and Catherine agree to marry. Olivier cuts the gloomy Epilogue lamenting Henry VI's failures.

Theatre directors have staged *Henry V* the play in a variety of ways, sometimes emphasizing the king's heroism, others his moral ambiguity. Movie directors have done the same, and comparing Olivier's production with later versions highlights its particular propagandistic nature. The most obvious comparison is with Kenneth Branagh's *Henry V* (1989), which lets viewers know they are in a different cinematic universe by staging the king's first appearance as an obvious allusion to Darth Vader. Critics often describe Branagh's production, made after the Vietnam and Falklands Wars, as anti-war, though it shares patriotic elements with Olivier's film. In any case, Branagh offers a more complex Henry, restoring such darker moments as the execution of the Southampton plotters, the threats before Harfleur, and the hanging of Bardolph, and making of the king a tortured hero at best. The difference between the two films is particularly stark with regard to the St. Crispin's Day speech and the battle of Agincourt itself. Olivier delivers the speech like a stage actor, proclaiming it as a rallying cry that might apply as easily to D-Day as to Agincourt, and doing so amid soldiers who are remarkably clean, well-dressed, and enthusiastic. Branagh's deliver is far more anguished, staring straight at the likely

---

38  The Folger Shakespeare Library version of the play is (Shakespeare 2004).

prospect of death, and he offers it to soldiers dirty, sodden from intense rain, and weary to the bone. As for the battle itself, Oliver's film follows the conventions of presenting warfare in the 1940s with a minimum of gore, while Branagh's extended scene often moves in slow motion to emphasize the brutality of war. While Branagh's renderings may be more honest, Olivier almost certainly came closer to what British citizens needed to hear and what the government wanted them to hear in 1944.[39]

## 12. Conclusions

In 1923, the anti-fascist artist Pablo Picasso observed, "Art is a lie that makes us realize truth." This seemingly paradoxical statement has relevance here. States most assuredly use art as propaganda, and—all other things being equal—more skillful art very likely makes better propaganda. By definition, propaganda is always biased, so effectively it is a lie or at least a distortion, but not all propaganda is equal, and the states that utilize it are not all morally equivalent. We might include among artists who lie to make us realize truth those who employ propaganda in pursuit of a larger goal that entails peace, prosperity, and justice for a greater number of people. Similarly, we might include among artists who lie but do not make us realize truth those who employ propaganda to promote inequality, intolerance, and violence. This is, of course, arbitrary, subjective, judgmental, and—in practical terms—absolutely necessary.

Some commentators now advocate assessing German films produced c.1933–1945 without considering Nazi filmmakers' intentions. Simultaneously, post-colonialist thinkers criticize films that justify British imperialism. This takes film scholarship in opposite directions and poses a complex problem. The best way to analyze a historical film is in terms of aesthetic quality, historicity, and presentism; therefore, it is impossible to ignore any filmmaker's agenda. Still, it is essential to remember that neither British imperialism nor British filmmaking in the 1930s and 1940s allow for black and white analysis. However, there is no such moral ambiguity in Nazism or the mad schemes of Adolf Hitler and his ilk. Moreover, while both states did distort the truth during World War II, they did so in service to profoundly different agendas. The world is eminently better off because of the Allied victory in 1945. To minimize the difference between Churchill's Britain and Hitler's Germany because the former was imperialistic takes relativism too far.[40]

The best art is that which speaks to the best in humanity and elicits the best behavior. Yet, even an aesthetically excellent historical film that moves people to good deeds is problematic if it presents inaccurate history. A good example should be a real example, and if there is any hope of humanity learning from history, we need an accurate understanding of it in all its nuances and subtlety. All historical films would be better if they paid more attention to historicity. Robert Brent Toplin, not unlike Picasso, has argued passionately that historically inaccurate films may still convey truth, and he is not altogether wrong. For example, the fictitious scene at the end of *Sophie's Choice*[41] probably evokes the horrors of the Holocaust as well as anything can. But with more and more people getting their 'history' from film and with the concurrent plague of partisan propaganda described as fake news, it is crucial for historians to point out inaccuracies and encourage critical thinking.[42]

Sometimes the excuse for inaccuracy is that real history is boring. Surely, no one studying Lancastrian and Tudor history believes that. A more serious concern is whether accurate history is even possible. The answer here is: "Lest we get caught up in postmodern angst about truth and the reality of the past, consider this: in the past certain conditions prevailed, certain events occurred, and certain people lived, acted, and died in certain ways. Much of the evidence for that is lost, but much survives. The truest account is the one that makes the best use of that evidence to give the most

---

[39] (*Henry V* 2015).
[40] For Picasso's comment (Barr 1939, p. 10); on the difference between Churchill and Hitler's approach to art (Cannadine 2018; Stewart 2019).
[41] (*Sophie's Choice* 1991).
[42] For example (Toplin 2010); see also (de Groot 2008; Hughes-Warrington 2006; Rosenstone 2018).

accurate description of that past." Informed reconstruction of the past is not the same as invention of the past. Good historians use a multiplicity of sources, approach them with critical skepticism and an awareness of how and why they were produced, and acknowledge when they speculate. They do not invent events like Michael Ingolby's mission to Spain or characters like Lord Wolfingham out of whole cloth, nor do any pretend to scientific certainty. But to acknowledge that complete accuracy is impossible is not to say that all accounts are equally accurate or inaccurate, and while historical accuracy may not interest film critics, there is no reason why it should not interest historians.[43]

Historicity aside, there are the related questions of a film's aesthetic appeal and effectiveness as propaganda. Once again, any assessment of either is bound to be subjective, as is all film criticism and indeed every individual reaction to a film. My own admittedly subjective, though reasonably well informed, judgment is that among the films discussed here, *Fire Over England* and *Henry V* are the most consistent in delivering pro-British propaganda, whereas *The Private Life of Henry VIII*, *Drake of England*, and *The Sea Hawk* sometimes get off-message. *Das Mädchen Johanna* is relentlessly anti-British (and anti-French), but it is not especially well made, even allowing for the deterioration of film and the absence of a digital upgrade. *Das Herz der Königin* is a better film and offers at least a germ of truth, but is profoundly skewed. By any reasonable standard the films made in Britain and America are aesthetically superior to those made in Germany, and to the twenty-first century viewer they seem more persuasive. They also received a more positive reaction from critics and performed better at the box office. That demonstrates the benefits of making historical films—even as propaganda—without Joseph Goebbels looking over your shoulder.

**Funding:** This research received no external funding.

**Conflicts of Interest:** The author declares no conflict of interest.

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
