# Peer review of "Lancastrians, Tudors, and World War II: British and German Historical Films as Propaganda, 1933–1945"

_arts, 1933_

Round 1
Reviewer 1 Report
I think the historical background in sections III and IV could be condensed a bit more. Keep the focus on where the films touch on the actual history and not give so much global historical background.
If there is a third relevant German film that could be discussed, that would increase the validity of the comparison. But given that the specialized time period of late medieval and Tudor England, there may not be another such film.
Problems with sentences:
p. 26, lines 10-11. Sentence about Raymond Massey has a missing verb.
p. 29, line 15. "a major [issue?]" missing word.
p. 31, line 18. Should it be "The Hymn of Leuthen" rather than "Hyman" ?
Author Response
I have corrected the errors the reviewer pointed out. I agree that the historical background in Sections III and IV needed to be condensed and focused more on where films touch on actual history, and I have revised those sections accordingly. I wish that I could comply with the reviewer’s recommendation to include a third relevant German film if one is available, but regrettably that is not the case.
Reviewer 2 Report
This article addresses the use of historical material as propaganda in films from Germany and Britain during World War II. By focusing specifically on films set in the late fifteenth and sixteenth century, and by bringing films made by both Germany and Britain into comparison, this article has the exciting potential to open fresh perspectives on the well-worn topic of propaganda in WWII films.
The introduction clearly sets out what the piece will cover, and provides a rationale for the selection of case studies. However, while I appreciate the need to gesture to the concept of the arts to overtly connect the piece to this journal’s specific concerns, the opening paragraph in particular felt a little generalised, and the introduction could perhaps be improved by a closer focus on the subject of propaganda in film, rather than on its broader history in the arts.
The article is strongest when dealing with historical material, which is unsurprising, given that the author makes clear that this is their area of expertise. I particularly appreciated the adept summary of the complexities of the history of the 15th/16th centuries. While this section (II) convincingly demonstrated why this period was significant for both Germany and Britain during WWII, it was difficult to see the relevance of all of the contextual information (especially in III and IV) and I would have liked to have seen the author connect some of the details raised in these sections more clearly to the analysis of the films themselves – for example, demonstrating how departures from the historical record support particular ideologies, and how those departures can be understood as having a propagandistic function.
The organisation of the films chronologically by release date allows them to be situated effectively within their particular contexts during WWII (which again are covered thoroughly) but the risk here is that the piece starts to read a little like a list, and the overall argument becomes rather lost. There is also a tendency throughout the discussion of the films to stray into description (particularly of plot), rather than analysis. The section on Henry V, for example, is primarily a list of events in the plot, with information of how these differ from the play, which I can’t help but feel is a missed opportunity in terms of analysing how this film functioned as propaganda. There is little to no engagement with the concept of the aesthetic composition of any of the films (no discussion of cinematography, for example) despite the fact that this is identified in the introduction as key to the overall argument. As a result, it is unfortunately difficult to read the assessment of the propaganda impact of these films as anything other than a subjective interpretation of their artistic worth. For example, despite the fact that Das Mädchen Johanna is dismissed as “not very good”, the author notes that it was still popular – suggesting in fact that it had the potential to be more influential than films that were arguably more refined (as a side-note, this kind of propaganda was exactly the kind that Goebbels favoured because it was unnoticeable). Similarly, evidence of examples from the press of the time would help to bolster the argument that comparisons were drawn between Hitler and Joan in this film (it is not always clear that this is the case). The most interesting points in the discussion of the films were raised by including parts of the dialogue from various films, but unfortunately much of this material was simply restated, and positioned as “obvious”, rather than explored in depth.
The issue with subjective assessment rather than analysis comes to a head in the conclusion, with the idea that “good” films make “better” propaganda – yet there is no evidence to substantiate this statement. Riefenstahl’s Triumph of the Will, for example, is a masterclass in documentary filmmaking and an aesthetically polished film, fitting the bill for a “good” (if morally reprehensible) film, but it was actually not that successful in terms of spreading propaganda in Germany itself, and is an example of the kind of strident propaganda that Goebbels strongly opposed. Similarly, the claim that all historical films would be “better” if they paid more attention to history is not only generalised, but also does not acknowledge the idea that historical films have a range of purposes and agendas that have nothing to do the notion of accuracy.
This article covers promising territory, and I would very much like to see it reworked with the following suggestions in mind:
- a closer focus on propaganda in film
- an explanation of the notion of “aesthetics” (or potentially abandoning this argument and concentrating instead on how and why this period is used in all the films)
- a more detailed analysis of the films
- an integration of historical context with the analysis of the films, which might require
- a restructure of the article to interweave some of the historical material more effectively within the analysis itself
- while there is evidence of extensive research here, I nonetheless would suggest that the author consider investigating more recent work on history in film. Rosenstone is of course important, but there are other, more recent theorists who might be useful (particularly in providing examples of how to conduct close analysis of films), such as Jerome de Groot’s Consuming History and Marnie Hughes-Warrington’s History Goes to the Movies, both of which might also allow for a more nuanced argument regarding the purposes and functions of the historical film.
Author Response
On 2 April 2002 I received email from Chloe Li advising me that my article ‘has been reviewed by experts in the field and we request that you make major revisions before it is processed further.’ However, the Review Reports actually varied significantly. Reviewer 1 recommended condensing the historical material in sections III and IV, Reviewer 2 recommended major revisions, and Reviewer 3 recommended publishing as is. The special issue editor, Andrew Nedd, requested that I ‘complete the revisions advised by reviewers 1 and 3 and take from reviewer 2 what you wish. In my opinion, the result would be an article that would be a fine contribution to the special issue.’ Therefore, I have proceeded on that basis.
Reviewer 2 says the article ‘has the exciting potential to open fresh perspectives on the well-worn topic of propaganda in WWII films’ but otherwise does not find much to like about it, saying the introduction and research design can be improved and the description of methods, presentation of results, and support for the conclusions must be improved. Per the reviewer’s suggestion, I have eliminated the first paragraph and condensed the historical material in Sections III and IV. However, with the other criticisms and suggestions, it is difficult to avoid the impression that the reviewer has taken the not uncommon path of objecting to my not having written the article the reviewer would have written, even though I indicated precisely what I intended to write and why in the first section of the article.
Reviewer 2 also objects to the subjective nature of my comments on the aesthetic appeal of the films I discuss. I acknowledge in the text that my comments are subjective, but I have now added the comment that all judgments about aesthetics are subjective. One can construct a list of seemingly objective criteria for judging direction, script, casting, acting, scenery, sound, cinematography, and so on, but each of those criteria will ultimately be based on a subjective judgment about what is good, bad, or mediocre. Even if film critics or film studies scholars agree about those criteria, they are still based on subjective judgments. And one need only read multiple reviews of a given film or multiple analyses by film studies scholars to find that disagreement about quality is common. Furthermore, propaganda, which this article address, is deliberately subjective, and the response to propaganda is subjective as well.
Reviewer 2 believes I should have done more to demonstrate ‘how departures from the historical record support particular ideologies, and how those departures can be understood as having a propagandistic function.’ The reviewer provides no examples of where I might have done so, but I would argue that I do exactly that throughout. Reviewer 2 objects to my ‘tendency throughout the discussion of the films to stray into description (particularly of plot) rather than analysis.’ But given that most of these films are relatively unknown to present-day readers and that plot is the point at which the films deviate from history for propagandistic purposes, there would not be much to analyze without some description. Reviewer 2 objects to my describing as ‘obvious’ rather than exploring in depth points that seem pretty obvious to me and that would not benefit significantly from belaboring the point. Reviewer 2 objects that there is no evidence in the text that comparisons were drawn between Joan and Hitler, but there is.
Reviewer 2 recommends a closer focus on propaganda, an explanation of aesthetics, and more detailed analysis of the films, with no hint of what this might entail. The reviewer also would prefer to see the historical material integrated with the analysis of the films. That is certainly one way to do it, and it is a method I have used elsewhere when it seemed appropriate. However, in this case, I believe that it would make the material unwieldly. The reviewer also has doubts about addressing history chronologically and then dealing with the films in order of release, claiming that makes it seem like a list. Well, it is a list, but it is a long, descriptive, analytical list.
Perhaps the most revealing recommendation is that I consider ‘investigating more recent work on history in film.’ The reviewer acknowledge that Rosenstone is ‘important,’ but recommends ‘more recent work by de Groot and Hughes-Warrington. In fact, Hughes-Warrington’s History Goes to the Movies appeared in 2006, de Groot’s Consuming History in 2008, and the third edition of Rosenstone’s History on Film/Film on History in 2018. It is hard to read as anything other than condescending the comment that these works ‘might be useful (particularly in providing examples of how to conduct close analysis of films)’ and ‘might allow a more nuanced argument regarding the purposes and functions of the historical film.’ But given the three scholars recommended, the implication seems to be that I should have adopted a postmodern approach that, as a historian, I do not find useful, as the article now explicitly notes.
Reviewer 3 Report
I really enjoyed reading this article. It offers an interesting and engaging addition to the topic of propaganda films - a topic I know far more about than fifteenth- and sixteenth-century England! The approach is innovative and any omissions in coverage have been well accounted for. I'm not sure I agree with the emphasis on accuracy, but that's just a difference in perspective. Claiming the main excuse is 'the real history is often more boring' (line 865) is, in my opinion, neglecting the numerous agendas of film-makers/stakeholders.
Overall I would recommend publishing as is, with a correction of minor errors. Those I noted:
Line 47 - I would change to the film industries in both nations were busy - otherwise this suggests more unity than is evident.
Line 91 - not rather than nor
Line 92 - are accurate (rather than is)
Line 527 - replaced
Line 598 - ends
Line 773 - film
Author Response
The errors Reviewer 3 pointed out have been corrected. Reviewer 3 generally praises the article, for which I am grateful. Reviewer 3 notes not being sure whether to agree with the emphasis on accuracy but attributes that, very reasonably, to a difference in perspective; however, I have provided a more detailed explanation for that claim. Reviewer 3 also questions the notion that inaccuracy results from the view that history is boring, which was somewhat tongue-in-cheek; however, I have modified that language as well.